# Immune System Modulations by Products of the Gut Microbiota

**DOI:** 10.3390/vaccines8030461

**Published:** 2020-08-21

**Authors:** Thierry Chénard, Karine Prévost, Jean Dubé, Eric Massé

**Affiliations:** Biochemistry and Functional Genomics Department, University of Sherbrooke, Sherbrooke, QC J1E 4K8, Canada; thierry.chenard@usherbrooke.ca (T.C.); Karine.Prevost@USherbrooke.ca (K.P.); Jean.Dube@USherbrooke.ca (J.D.)

**Keywords:** immune system, microbiota, short-chain fatty acids, microbe-associated molecular patterns

## Abstract

The gut microbiota, which consists of all bacteria, viruses, fungus, and protozoa living in the intestine, and the immune system have co-evolved in a symbiotic relationship since the origin of the immune system. The bacterial community forming the microbiota plays an important role in the regulation of multiple aspects of the immune system. This regulation depends, among other things, on the production of a variety of metabolites by the microbiota. These metabolites range from small molecules to large macro-molecules. All types of immune cells from the host interact with these metabolites resulting in the activation of different pathways, which result in either positive or negative responses. The understanding of these pathways and their modulations will help establish the microbiota as a therapeutic target in the prevention and treatment of a variety of immune-related diseases.

## 1. Introduction

The microbiota and the immune system have co-evolved as parts of a symbiotic relationship, probably since the origin of the innate immune system, more than 700 million years ago [1]. While the microbiota plays an important role in the induction, training, and regulation of the immune response, the immune system developed to distinguish between commensal and pathogenic bacteria [2]. Interactions between the microbiota and the immune system depend on a variety of metabolites and receptors activating and inhibiting different signaling pathways in the host cells, and can be both beneficial and detrimental to the host health. Metabolites involved in these interactions range from simple metabolic byproducts such as short chain fatty acids (SCFA) to complex structural macromolecules necessary for bacterial integrity, such as peptidoglycan and lipopolysaccharides (LPS). The abundance of these metabolites are linked to the bacterial composition of the microbiome, which can be affected both by the diet and by environmental factors, with the specific effects still being investigated [3,4]. Some of the mechanisms involved in the interactions between the microbiome and the host’s cells are discussed in this review.

## 2. Metabolites Originating from the Microbiome with Effects on the Innate Immune System

The innate immune system consists of elements such as the complement system, Toll-like receptors (TLR), other pattern recognition receptors (PRR), and phagocytic cells, which have the capacity to recognize and destroy the nonself, but lack the specificity to differentiate between helpful and harmful elements [1]. Additionally, this lack of specificity in the targeting can lead to the immune system to damage surrounding tissues or even the entire organism, due to the activation of inflammatory and sepsis pathways. This is especially true when the regulation of the innate immune system is perturbed in the context of disease and dysbiosis [5]. PRR are divided into four different family TLRs, nucleotide-binding oligomerization domain (NOD)-Leucin Rich Repeats-containing receptors (NLR), the retinoic acid-inducible gene 1-like receptors (RLR), and the C-type lectin receptors (CLR). The activation of the various PRRs lead to distinct expression patterns of proteins, such as mostly proinflammatory cytokines, antimicrobial proteins, and chemokines, resulting in the induction of infected cell death, the co-stimulation of the adaptive immune system, and inflammation at the site of infection [6,7]. All PRRs are able to recognize molecular patterns invading pathogens and self-components, but each family is specialized in recognition of patterns from specific groups. RLRs and CLRs are responsible for the detection of ligands of viral and fungal origins, respectively, and are not discussed in this review. NLRs detect ligands of bacterial origins and TLRs have the ability to detect ligands of all origins.

The intestinal epithelium consists of a monolayer of polarized cells, creating a border between the intestinal lumen, where the microbiota is located, and the rest of the body. In part, because they do not originate from hematopoietic cells like the other cell types involved in the immune system, intestinal epithelial cells are not usually considered to be a part of the innate immune system. However, intestinal epithelial cells possess multiple receptors associated with innate immunity, such as TLR2 and TLR4. These cells also have the capacity to affect the microbiome directly through the production of mucus and the release of antimicrobial peptides into the intestinal lumen [8]. As the intestinal epithelium is located at the interface between the intestinal microbiota and the host tissues, this makes it an important component of the crosstalk between the innate immune system and the intestinal microbiota [9]. A vast array of molecules originating from various bacteria present in the microbiome are shown to influence the innate immune system.

### 2.1. Short-Chain Fatty Acids

SCFA, fatty acids with less than six carbon atoms, are byproducts of the saccharolytic fermentation (converting hexoses to pyruvate) of non-digestible carbohydrates through various bacteria of the microbiome, and have a large number of potential positive effects on the hosts immune system [10]. The three primary SCFA produced by the microbiome are acetate, propionate, and butyrate, with approximate molar proportions of 3:1:1 [11]. Each of these SCFAs are metabolized differently and activate distinct signaling pathways. While most SCFA have a positive impact on health, butyrate is the major SCFA affecting the immune system. There is a plethora of bacteria in the intestine that have the capacity to generate butyrate, but the main producers belong to the phylum Firmicutes, particularly *Faecalibacterium parasitizes*, *Clostridium leptum, Eubacterium rectale*, and species of the *Roseburia* genus [12]. Selected members of other phyla such as Actinobacteria, Bacterizes, Fusobacteria, Proteobacteria, Spirochaetes, and Hermitages also have the potential to produce butyrate under certain conditions, using different synthesis pathways [13]. Propionate and acetate can be produced by species like *Ackermann’s muciniphila* during mucin fermentation [14], or through carbohydrate fermentation in bacteria, such as the *Bifidobacterium* species [15].

Propionate generated in the gut is mostly transported and sequestered to the liver, where it is either oxidized or used as a substrate for gluconeogenesis [16]. Propionate acts through the activation of the G protein-coupled receptor (GPR)41, also known as the free fatty acid receptor 3 (FFAR3) in the peripheral nervous system, which leads to increased abundance of glucose-6-phosphatase (G6Pase) in the jejunum [17]. This increase in intestinal gluconeogenesis is associated with increased glucose and insulin sensitivity, which helps prevent type 2 diabetes. The activation of GPR41 in sympathetic neurons reduces the intracellular concentration of cAMP and activates the extracellular signal-regulated kinase (ERK) pathway, by promoting the phosphorylation of ERK1/2, leading to further protein and transcription responses [18].

Butyrate mainly serves as the major source of energy for colonocytes [19]. Together with propionate, butyrate can also induce intestinal gluconeogenesis and serve as a component of this pathway. Moreover, butyrate can directly induce the gluconeogenesis-associated genes via an increase in cyclic adenosine monophosphate (cAMP).

Acetate is converted to acetyl-coenzyme A (CoA), a central metabolite serving as an intermediate between glycolysis and the tricarboxylic acid (TCA) cycle. Acetate is also used as a metabolite in many other synthesis pathways, such as the ones for sterols and ketones, and has signaling effects by affecting histone acetylation, in peripheric tissues, via the activity of acetyl-CoA synthetase (ACSS) 1 and 2 [20].

#### 2.1.1. The Effect of SCFA on Intestinal Epithelial Cells

The intestine is the organ that possesses the highest amounts of immune activity in the body, as it contains multiple inductor (Peyer’s patches, lymph nodes, and lymphoid follicles) and effector (epithelium and the lamina propria) sites [21]. Each of these sites and the various cells composing them possess different functions and activities that affect the overall immune response of the intestine, some of which can be affected by butyrate. The mucosal layer is the first line of defense against microbes in the intestine, it is produced by the goblet cells of the epithelium and its production can be influenced by various physiological and immune mediators [22]. Mucin 2 (MUC-2) is the most prominent mucin secreted by these cells, and butyrate was shown to increase its expression by increasing the production of prostaglandin (PG) in subepithelial myofibroblasts located next to the intestinal epithelium [23].

#### 2.1.2. The Effect of SCFA on Inflammation and Tight Junctions

Multiple studies have shown that increased butyrate availability in the intestine results in the activation of anti-inflammatory pathways and the inhibition of pro-inflammatory pathways [24]. Some of the pathways affected include a downregulation of the expression of TLR4 [25], which is linked to the production of pro-inflammatory cytokines and the activation of the nuclear factor kappa-light-chain-enhancer of the activated B cells (NF-κB) [26]. The downregulation of NF-κB by butyrate also involves the peroxisome-proliferator-activated-receptor γ (PPARγ) and the vitamin D receptor (VDR). This results in attenuation of the tumor-necrosis-factor α (TNF-α) and LPS-dependent activation of NF-κB [27]. Butyrate also influences the assembly of tight junction complexes, by increasing AMP-activated protein kinase (AMPK) activity, but not through increased expression of the main tight junction proteins [28]. The combined effect of butyrate on the formation of tight junctions and the production of MUC-2, indicate an important role of butyrate in promoting the integrity of the barrier function of the intestinal epithelium.

Tight junctions are important in the regulation of elements, including bacteria and toxins across the intestinal epithelium into the lamina propria, and is disrupted in a variety of diseases linked to inflammation and auto-immunity [29]. It was shown that increased gut permeability resulting from altered tight junction protein expression in germ-free mice is reflected in the increased permeability of the blood–brain-barrier (BBB) [30]. The permeability of the BBB decreases in germ free mice exposed to a normal gut flora or after oral gavage with sodium butyrate. Similar effects were seen on the permeability of the blood–testis-barrier (BTB) in germ-free mice, after exposure to the bacteria *Clostridium tyrobutylicum*, which produces large amounts of butyrate [31]. Increased permeability at the BBB and BTB results in augmented infiltration of inflammatory molecules and immune cells to the brain and testis, often found in patients suffering from neurological diseases like multiple sclerosis or Alzheimer’s [32,33].

Finally, butyrate prevents neutrophil and eosinophil infiltration to the mucosal layer and decreases the number of activated B lymphocytes in the cecal lymph nodes, as well as the migration of macrophages to those nodes, leading to a reduction in the inflammatory profile of the intestinal mucosa [34]. Neutrophils and eosinophils are varieties of white blood cells, with an important role in inflammation and the innate immune response. However, they are also known to activate the adaptive immune response, due to their potential to process and present antigen. Moreover, eosinophils and neutrophils produce chemokines, cytokines, and granular content, activating both B and T cells proliferation and activation, as well as motility and infiltration [35,36,37,38]. More information on the activation and activity of B and T cells are presented in the section on adaptive immunity. An overview of the actions of butyrate on the innate immune system is available in Figure 1.

### 2.2. Microbe-Associated Molecular Patterns and Pathogen-Associated Molecular Patterns

Both commensal and pathogenic bacteria present in the intestine have the capacity to produce a variety of molecules that can be recognized by PRRs [39]. These microbe- or pathogen-associated molecular patterns (MAMPs and PAMPs) include, but are not limited to, bacterial LPS, peptidoglycan, lipoteichoic acid (LTA), toxins, and flagellin. Since commensal microorganisms, not only pathogens, have the capacity to produce these molecules, the term MAMPs is preferred to PAMPs [40]. The interactions between PRRs and MAMPs, trigger signaling cascades and modifications in expression, resulting in the modulation of the activation and function of the innate immune response, including the release of interferons and other cytokines.

#### 2.2.1. Lipopolysaccharides

LPS is the main component of the Gram-negative bacteria outer membrane and can be divided into three parts—the lipid A, which is the most conserved part; a core oligosaccharide chain; and the O-antigen, which is a variable polysaccharide chain [41]. The composition of the lipid A part of the LPS is in part responsible for activating the immune system. Indeed, both additional and longer acetylation of the lipid A reduces its potential to activate the immune system [42]. LPS is recognized in the extracellular space by TLR4, which has the ability to recognize other molecules, such as the teichuronic acid from Gram-positive bacteria, the F protein of respiratory syncytial virus, or even some human heat shock proteins [26]. TLRs, which belong to the PRR family, detect the presence of certain molecules, and initiate the immediate response of the innate immune system and the long-lasting adaptive immune response. TLR4 signaling is extensively reported in multiple recent reviews [43,44,45]. In short, TLR4 associates with the myeloid differentiation 2 protein (MD2) on the cell surface, by interacting with LPS. Afterward, dimerization of TLR4/MD2 complexes occurs, which induces conformational changes to the structure of TLR4, exposing the Toll/interleukin-1 receptor-like (TIR) domain of their intracellular tails. These domains are then able to interact with other TIR domains present on four distinct adaptor proteins; the myeloid differentiation primary response protein 88 (MyD88), MyD88-adaptor like (MAL), TIR domain-containing adaptor inducing interferon-β (TRIF), and TRIF-related adaptor molecule (TRAM). The MyD88/MAL pathway is responsible for the early activation of NF-κB. It activates a cascade of kinases, resulting in the phosphorylation of the inhibitor of kappa B (IκB), leading to the translocation to the nucleus of NF-κB and its activation. Activated NF-κB binds to enhancer elements of the immunoglobulin (Ig) kappa light-chain of activated B cells (κB sites), resulting in the increased expression of various pro-inflammatory genes. This also stimulates the release of major inflammatory cytokines, such as interleukin (IL)-1β, IL-6, and TNF-α [46]. The TRIF/TRAM pathway is activated in the endosomal compartment, after the internalization of the activated TLR4/MD2 complex dimer, and results in the activation of transcription factor interferon regulatory factor 3 (IRF3). IRF3 activation leads to the expression of type 1 interferon (IFN) and IFN-inducible chemokines, such as IL-10. Type 1 IFN is a pleiotropic family of cytokines, critical in mediating the immune response to various stimulants in the intestine [47]. Type 1 IFN, in most cases, is important in maintaining gut homeostasis and protecting against inflammation, but is also involved in the induction or support of pro-inflammatory responses. The TRIF/TRAM pathway also activates TNF-α production, resulting in late phase NF-κB activation. An overview of the effects of LPS on immunity is available as part of Figure 2.

#### 2.2.2. Bacterial Peptidoglycan

Peptidoglycan is a large polymeric molecule found in the cell wall of all bacteria. The general structure is conserved between bacterial species, but differences in the composition and length of the backbone and the crosslinking, increase the variability of the peptidoglycan. Additionally, there can be differences in modifications made on the sugar structure (deacetylation, O-acetylation, and N-glycosylation) between bacteria, which affect the properties of the cell wall and the pathogenesis of the bacteria. The innate immune system can detect peptidoglycan or its fragments using different receptors, peptidoglycan recognition proteins (PGLYRPs), TLRs, and NLR [48]. PGLYRPs consists of four soluble proteins (PGLYRP1 to 4) that bind to peptidoglycan [49]. PGLYRP1 is mostly expressed by eosinophils and neutrophils, and could potentially directly act on inflammation. Whereas PGLYRP2 is produced in the liver and liberated into the circulation, both PGLYRP3 and 4 are mainly expressed by mucosal surfaces, skin keratocytes, and other surface cells. PGLYRP1, 3 and 4 can induce bacterial cell death directly, while PGLYRP2 shows an amidase activity and hydrolyzes the peptidoglycan of bacterial cell walls. PGLYRPs can induce bacterial death due to cooperation with PRRs, leading to increased phagocytosis. Both PGLYRP3 and 4 activation by peptidoglycan have anti-inflammatory effects and were shown to affect the composition of the gut microbiota [50,51]. PGLYRP2 plays a pro-inflammatory role in peptidoglycan-induced inflammation, while PGLYRP1 plays an anti-inflammatory role in the same circumstance [52].

The potential of peptidoglycan to activate TLR2 is still highly debated. Historically, it was thought that peptidoglycan was a stimulator of TLR2 but improved purification showed the main activators to be diacetylated and triacetylated lipoproteins that are linked to peptidoglycan [53,54]. TLR2 form heterodimers with TLR1 or TLR6 and, when bound to its ligand, activates the same MyD88 pathway as TLR4, leading to the activation of NF-κB, and the release of pro-inflammatory cytokines [55].

The final group of peptidoglycan sensors are the NLRs, NOD1, and NOD2, which are both expressed within the intestinal epithelium and in various intestinal immune cells [56]. NOD1 and 2 both detect intracellular peptidoglycan and peptidoglycan fragments, with different preferences. NOD1 recognizes part of the peptidoglycan found mostly in Gram-negative bacteria and NOD2 binds a dipeptide found in the peptidoglycan of most bacteria. After the binding of their respective ligand, both NOD1 and 2 self-oligomerize. Following this, they interact with receptor-interacting serine/threonine protein kinase 2 (RIPK2), leading to the activation of two different pathways—(1) activation of NF-κB and (2) activation of mitogen-activated protein kinase kinase kinase 7 (MAP3K7) and the mitogen-activated protein kinase (MAPK) signaling cascade. Both of these pathways lead to the production of inflammatory cytokines and chemokines [48]. Additionally, NOD1 and 2 were demonstrated to induce expression of type-1 IFN genes during bacterial infection and only NOD2 had this potential during infections with single-stranded RNA viruses, which could have anti-inflammatory effects. NOD1 and 2 also showed effects on the adaptive immune system and is discussed later. The variety of sensors able to detect peptidoglycan and either their anti- or pro-inflammatory effects are indicative of the importance of the location and the context of the detection of peptidoglycan by the innate immune system, on the precise response of the system. An overview of the effects of peptidoglycan on immunity is available in Figure 2.

#### 2.2.3. Lipoteichoic Acid

LTA is a component of the cell wall that is unique to Gram-positive bacteria, such as *Bacillus subtilis*, *Staphylococcus aureus*, or *Listeria monocytogenes*. LTA consists of a glycerol phosphate backbone attached to the membrane with a glycolipid anchor and the exact structure varies considerably between species [57]. Just like peptidoglycan, LTA is considered an agonist of TLR2, but some studies indicated that the effects from LTA originated from contamination of the commercial preparations by lipoproteins or lipopeptides [58,59]. Additionally, the LTA of some bacterial species such as *Clostridium butyricum* or *Lactobacillus plantarum* are instead able to inhibit the inflammatory response induced by other PAMPs [53,57,60]. These varying effects resulting from different LTAs indicate that specific LTA structures can activate different pathways, leading to opposing effects. An overview of the effects of LTA on immunity is available in Figure 2.

#### 2.2.4. Flagellin

Flagellin is the main structural component of the flagella of mobile bacteria and is being studied as an adjuvant to boost the immune response in poorly immunogenic vaccines [61]. Flagellin is an agonist of TLR5 which, when activated, forms a homodimer and stimulates the same MyD88-dependent pathway activated by TLR2 and TLR4. Otherwise, active TLR5 can also form a heterodimer with TLR4, to activate the TRIF/TRAM pathway that was earlier described to be activated by the TLR4 homodimer in the presence of LPS. Certain Gram-negative bacteria, such as *Salmonella*, have a type III secretory system that transfer their flagellin and other effector proteins to the host cell cytoplasm. The NLR family, apoptosis inhibitory proteins (NAIP), NAIP5, and 6, then senses intracellular flagellin. The binding of flagellin to NAIP5 and 6 leads to the activation of caspase-1 and to the production of active IL-1β and IL-18 [62], which are pro-inflammatory cytokines that are important for the host-defense responses to infections. Remarkably, treatment with flagellin was also shown to have the potential to protect against pathogens, radiations, and more, due to its activation of the immune system [61]. An overview of the effects of flagellin on immunity is available in Figure 2.

#### 2.2.5. Toxins

Multiple bacterial pathogens have the capacity to produce a diversity of toxic compounds that can interact with the host cells and potentially with the immune system. Some of the most studied toxins are those produced by *Clostridium difficile.* It produces 2 different exotoxins, toxin A and B (TcdA and TcdB, respectively), which belong to the large clostridial toxin family, along with toxins from *Clostridum novyi*, *Clostridium perfringens*, and *Clostridium sordelli* [63]. TcdA and B are classified as AB toxins where the B subunit is responsible for the delivery of the A subunit to the host’s cytosol, where it can produce its effect. *Bacillus anthracis* and other pathogens also produce AB toxins that affect the host similarly [64,65]. TcdA and B bind to different glycosylated receptors on the host cell, TcdA binds to sucrase-isomaltase (SI) and glycoprotein 96 (gp96), while TcdB is known to bind to chondroitin sulfate proteoglycan 4 (CSPG4), as well as poliovirus receptor-like protein (PVRL3), and frizzled proteins 1, 2, and 7. All of these receptors are expressed by different cells at the epithelium and subepithelium of the intestine. TcdA and B use different endocytic pathways to enter their target cells. TcdA uses protein kinase C and casein kinase substrate in neuron protein 2 (PACSIN2)-mediated transport, while TcdB depends on the clathrin-mediated endocytic pathway. Both TcdA and B affect the host cell in various ways. In the endosome, under low pH, they incur a conformational change that exposes a hydrophobic region that inserts itself in eukaryotic membranes to form pores that translocate their active domain to the cytosol [66]. This active domain is a glucosyltransferase domain that targets and inactivates the RHO family GTPases (RHO, Ras-related C3 botulinum toxin substrate 1 (RAC1), and cell division control 42 (CDC42)). The modification of these proteins lead to modifications of the cytoskeleton and apoptosis. TcdB can also promote the formation of the NADPH oxidase complex on endosomes, which increases the production of reactive oxygen species, leading to necrosis. The cytosolic presence of the toxins can also be sensed by MAPK and the pyrin inflammasome, leading to increased secretion of IL-1β, IL-8, TNF-α, and IL-6 [67]. These mechanisms lead to increased inflammation, disruption of tight junctions, and tissue damage linked to the activation of innate immune system.

Moreover, certain strains of *C. difficile* can produce a third toxin named *C. difficile* transferase (CDT), an actin-specific ADP-ribosyltransferase composed of two subunits (CdtA and CdtB), correlating with increased mortality and morbidity during infections. This binary toxin affects cells differently than TcdA and B, after using a similar entryway [67]. The CdtB subunit binds to the lipolysis-stimulated lipoprotein receptor (LSR) of colonic cells, under the form of a heptamer. Afterward, the CdtA subunit binds to CdtB, enabling the endocytosis of the complex. Once inside the acidified endosome, CdtB inserts itself into the membrane, allowing CdtA to enter the cytosol. In the cytosol, CdtA can collapse the cytoskeleton and creates microtubule protrusions, thereby increasing adhesion between *C. difficile* and the host cells. Additionally, CDT can be recognized by TLR2 and cause apoptosis of eosinophils, contributing to overall inflammation associated with infections.

#### 2.2.6. Tryptophan-Derived Metabolites

Tryptophan is an essential amino acid in humans, meaning the body cannot produce it. It has been shown that tryptophan helps reduce inflammation and protect against colitis by modulating the activation of the immune system [68]. Most of the consumed tryptophan is used in protein synthesis, while a small part of it is metabolized either by the host’s cells (producing kynurenines, serotonin, and melatonin), the microbiota (resulting in the production of indole, indolic acid derivatives, and tryptamine), or even the digestive environment (producing indole-3-methanol, indole-3-acetonitrile that gets converted to glucoside derivatives by the acidity of the gastric juice) [69]. These tryptophan derived metabolites have the capacity to interact with the aryl hydrocarbon receptor (*AhR*), which modifies the expression of downstream genes and affect both the innate and adaptive immune response. Moreover, the bacteria-derived metabolites are also known to activate the pregnane X receptor (PXR) [70]. We will specifically discuss the activity of microbiota-derived metabolites below.

Indole, one of the major metabolites resulting from the metabolism of tryptophan, is found to fortify tight junctions between the intestinal epithelium cells, through its binding to PXR [71]. PXR seems to reduce TLR4 activation, since both receptors and indole-secreting bacteria need to be present at appropriate levels, to prevent intestinal barrier dysfunction. Indole binding to *AhR* is linked to the local production of IL-22, which protects against pathogenic infections. Additionally, indole is linked to reduced activation of NF-κB dependent on TNF-α, increased expression of the anti-inflammatory cytokine IL-10, and reduction of IL-8 expression, but the exact mechanisms are not elucidated [72].

Two different indolic acid derivatives (indolyl propionic acid and indolyl acrylic acid) were described as having the capacity to enhance IL-10 production, following LPS stimulation [71,73]. Skatole, also known as 3-methylindole, is another indolic-acid-derived metabolite with effects on *AhR* signaling, with an efficacy comparable to indole. However, it is usually present in the intestine in low concentrations, possibly because its synthesis depends on a two-step production pathway, mediated by two different bacteria. Skatole is identified as a pneumotoxin in ruminants, but to date no such effect was detected in humans and mice [74]. It was also described as an inhibitor and a substrate of the mitochondrial cholesterol side-chain cleavage enzyme (CYP11A1) [75]. Presence of skatole leads to increased inflammation, associated with the irritable bowel syndrome, through a decrease in glucocorticoid production, instead of through interactions with the immune system. Finally, it had bacteriostatic effects on the growth of gram-negative enterobacteria. All this information indicated that skatole could have both pro and anti-inflammatory effects in the intestine, possibly depending on its concentration and location.

Tryptamine is a metabolite resulting from the decarboxylation of tryptophan, which occurs only in some specific bacteria such as *Clostridium sporogenes* and *Ruminococcus gnavus* [76]. The Human Microbiome Project showed that at least 10% of microbiome samples contained at least one gene responsible for tryptophan decarboxylation. Like other tryptophan metabolites, tryptamine had the capacity to activate *AhR* in the intestine to release IL-22 [70]. *AhR* also plays a role in regulating tryptamine production, since *AhR*-knockout mice showed reduced levels of tryptamine after dextran sodium sulfate (DSS)-treatment. Tryptamine also has additional functions in maintaining intestinal homeostasis, without a direct effect on the innate immune system, such as, inducing the release of serotonin and increasing its inhibitory effect or inducing the release of ions through intestinal epithelial cells [68]. While many of the AhR-activating molecules originate from tryptophan metabolism, butyrate also induces *AhR* activity, leading to the protective effects of *AhR* [77].

## 3. Metabolites Originating from the Microbiome with Effects on the Adaptive Immune System and Vaccination

The adaptive immune system is responsible for the long-term memory of the immune system, and vaccination depends on this memory. The main cell types involved in the adaptive immune system are the B and T lymphocytes and their more specialized subsets with different specific activities [1]. T lymphocytes usually originate from the thymus, while B lymphocytes originate from the bone marrow of adults and the liver of fetuses. Both B and T lymphocytes can produce a large diversity of antigen receptors-based surface receptors, T lymphocytes contain T-cell receptors (TCR) for antigen, while B lymphocytes contain Ig receptors. All of these are specialized in recognizing nonself molecules, referred to as antigens, with high specificity. The high diversity of these receptors is not encoded directly in the genes but results from a recombination of a finite set of adjacent genes during development. T lymphocyte are separated into effector (or cytotoxic) T cells (Teff), helper T cells, regulatory T cells (Treg), memory T cells, and more [78]. B lymphocytes can be separated into plasmablast, plasma cell, regulatory B cells (Breg), memory B cells, and more [79]. The activation of Teff and helper T cells is dependent on their interaction with antigen-presenting cells, such as dendritic cells [80]. The helper T cells assists in the activation of other lymphocytes (including B cells), through the production of various cytokines, while Teff destroy infected cells after they recognize an antigen presented by the major histocompatibility class (MHC) 1 protein, at the target cell surface. The activation of B lymphocytes happens in secondary lymphoid organs, such as the spleen and lymph nodes, and depend either on an interaction with helper T cells or T cell-independent antigens [81]. Plasmablasts are B lymphocytes that secrete antibodies, while being able to divide. Plasma cells, B lymphocytes that produce antigen-specific antibodies, can be short- or long-lived, depending on whether they are produced early (short-lived) or later (long-lived and originating from memory B cells) during the immune response. Treg and Breg are responsible for controlling other lymphocytes to prevent uncontrolled or continuous activation of the immune system. When they are activated, by binding of an antigen to their receptors, both types of lymphocytes undergo an explosive clonal proliferation and differentiation, ending with the death of the most involved antigen-specific lymphocytes. The remaining living few become long-lived B and T memory cells that are responsible for a quicker and more efficient response in case of reinfection, which is the main mechanism behind vaccination. Many factors affect the immune response to vaccination, including the type and composition of the vaccine (dosage, adjuvants, nature of the antigen, administration route, etc.), as well as the person’s age and health.

### 3.1. SCFA Effect on Regulatory T Cells

In addition to the effects of SCFA on the innate immune system, butyrate and propionate (but not acetate) were shown to increase extrathymic conserved non-coding sequence 1 (CNS1)-dependent differentiation and proliferation of Treg cells, in mice [82,83]. Treg are immunosuppressive and usually downregulate the induction and proliferation of other T cells. Treg cells expressing the transcription factor forkhead box P3 (*Foxp3*), play a key role in the regulation of intestinal inflammation. CNS1 is an intronic *Foxp3* enhancer that is essential for extrathymic Treg cell differentiation but not thymic Treg cell differentiation. Treatment of colonic Treg with SCFA specifically increased IL-10 expression and increased the suppressive capacity of the colonic Treg towards the pro-inflammatory CD4^+^ Teff [83]. Additionally, expression levels of *GPR15*, whose role consists in the migration and retention of Treg cells to the large intestine [84], were increased, resulting in the augmented presence of Treg in the colon lumen. Finally, propionate only affects Treg proliferation and activity when the free fatty acid receptor 2 (Ffar2 or GPR43) is present, and showed a reduced expression of histone deacetylase (HDAC)6 and 9, which could explain the vast effect of SCFA on the expression levels in Treg. An additional effect resulting from the inhibition of HDAC and modifications of gene expression by butyrate is an increased proliferation of the colonic epithelium in the bottom half of the crypts, while increasing apoptosis in the cells exfoliated to the lumen, but inhibits this same proliferation in cancerous cells, via the Warburg effect [85].

### 3.2. Peptidoglycan Effect on Antigen Presentation

As described above, both NOD1 and 2 can detect peptidoglycan fragments in circulation, to activate the innate immune system, but they also block responses from the adaptive immune system. Nod1/2 KO mice were more susceptible to reinfections with *Bacillus anthracis*, probably due to failure to produce enough antibodies from memory cells [86]. The exact mechanisms behind the effect of NOD1 and 2 on the production of antibodies are not clear but might involve impaired production of chemokine ligand 5 (CCL5) and IL-12p70, in the absence of NOD1 and 2. CCL5 is involved in the recruitment of immune cells to the site of infection, and IL-12p70 induces the differentiation of helper T cells into proinflammatory Th1 cells. Derivatives of peptidoglycan were identified as acting as an adjuvant in the production of IgG, resulting from Th2 polarization [56]. Finally, NOD1 and 2 functions in T cells were important in activating ERK signaling, the selection of thymic Teff and limiting Th17 response. All these elements indicated that peptidoglycan-dependent activation of NOD1 and 2 is important in the activation of T cells and in developing the memory aspect of the immune system, even if the mechanisms are not fully elucidated.

### 3.3. Flagellin

Flagellin is being investigated as having a significant potential as a natural adjuvant in vaccination. A study showed that TLR5-deficient mice, incapable of responding to flagellin, could not obtain a protective immunity, following the administration of a nonadjuvanted subunit vaccine for influenza [87]. The same effect were seen in germ-free mice and antibiotic-treated mice, in which the antibody response was saved by the administration of a flagellated strain of *Escherichia coli*, suggesting the importance of flagellin from the microbiome on efficient vaccine response [88]. The exact mechanisms by which flagellin and TLR5 affect the production of antibodies are partially defined. It is dependent on the production of proinflammatory cytokines and chemokines stimulated by flagellin, helping in the recruitment of lymphocytes and the encounters between antigen and receptors. TLR5 and the NAIP-inflammasome expressed by B and T lymphocytes are in part responsible for the stimulation of antibody production by these cells, after stimulation by flagellin. However, a study showed that a MyD88 independent pathway was also involved in flagellin recognition and activation of the immune system [89].

### 3.4. Bacterial Effects on the Efficiency of Vaccination

Only a small number of small clinical studies investigated the link between the composition of the microbiome and the response to vaccines [90]. These studies indicate that some bacterial species vary between high and low responders. Results showed that high levels of *Bifidobacterium longum* and *Streptococcus gallolyticus* correlated with higher response to vaccine. Moreover, high abundance of bacteria from the phylum Bacteroidetes and phylum Proteobacteria, order Enterobacteriales and Pseudomonadales, were associated with lower response to vaccination. Multiple studies implicated the microbiota in the effective response of the adaptive immune system during vaccination, even though the exact mechanism is still not completely understood.

## 4. Other Potential Metabolites and Perspectives

In addition to the different metabolites mentioned previously, other metabolites such as indoles, 4-cresol, branched-chain amino acids, deoxycholic acid, and trimethylamine, produced by the microbiota were shown to affect the development of cardiovascular diseases, obesity, fatty liver disease, atherosclerosis, colorectal cancer, and type 2 diabetes, without known interaction with the immune system [91,92,93,94,95]. Although some of these metabolites (i.e., branched-chain amino acids) are known to have immunomodulatory capacities, their mechanisms are not fully understood. Still, these metabolites characterized mechanisms in other cell types (i.e., adipocytes and cardiomyocytes), associating them with the development of their associated diseases. Further studies on these metabolites and their effects on health and the immune system could further help establish the microbiota as a potential target in the treatment of diseases. The understanding of the mechanisms linking the microbiota to the activation of the immune system could be helpful in establishing the microbiome as a key player in personalized medicine, through the manipulation of its composition.

## 5. Conclusions

In conclusion, multiple pathways were discovered through which the microbiome could affect both the innate and adaptive immune system, either in a positive or in a negative fashion. The continued investigation of these pathways through the perspective of the microbiome showed how symbiotic bacterial populations are critical to the development of many diseases. Moreover, these investigations on the microbiome will help in the eventual establishment of the microbiome as a therapeutic tool in personalized medicine.

## Figures and Tables

**Figure 1 vaccines-08-00461-f001:**
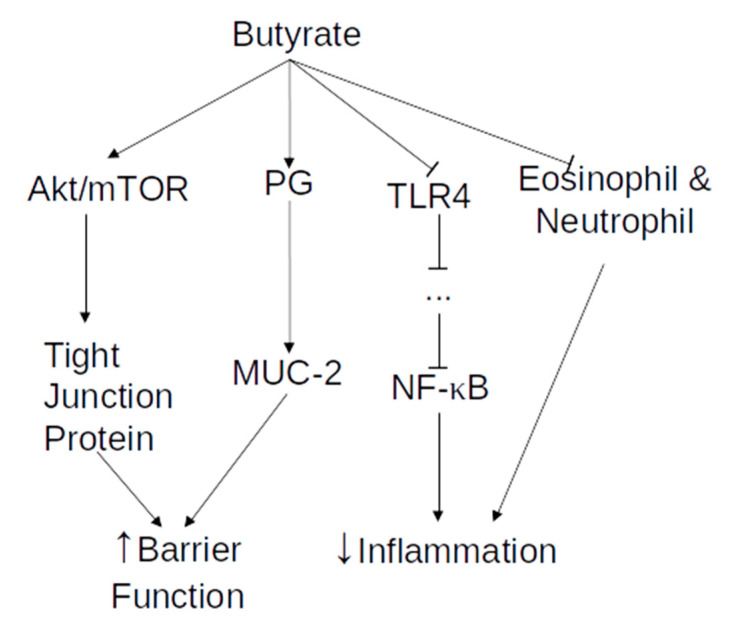
Overview of butyrate effects on the innate immune system. Arrow end—increase in activity or concentration. Blunt end—decrease in activity or concentration.

**Figure 2 vaccines-08-00461-f002:**
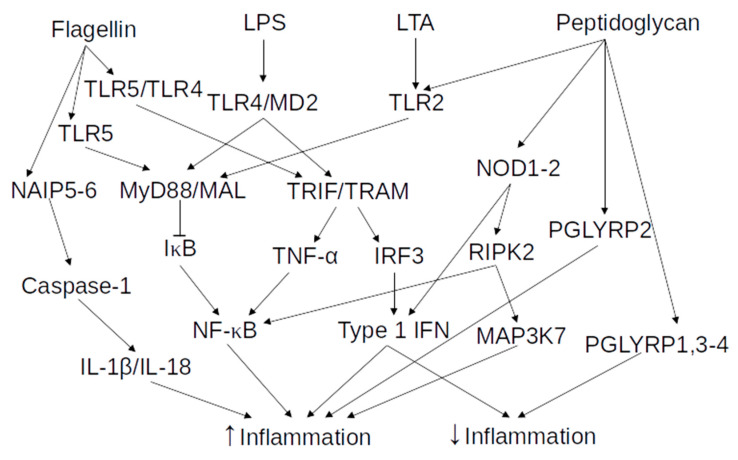
Overview of the selected MAMPs effects on the innate immune system. Arrow end—increase in activity or concentration. Blunt end—decrease in activity or concentration.

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
