# Peer review of "Immune System Modulations by Products of the Gut Microbiota"

_vaccines, 2020, doi:10.3390/vaccines8030461_

Round 1

Reviewer 1 Report

Its a well-written, comprehensive review with importance to the field about microbial productions and their contribution to aspects of innate and adaptive immunity.

Important points:

  • I am not quite sure whether this article topic fits into the scope of "Vaccines" Journal
  • Suggest including a table summarizing the microbial molecules, their effects on immune system and related pathways, and the respective references
  • Divide the paper clearly into "innate" and "adaptive" aspects, and put microbial components as sub-heading; e.g.
    • 1. Intro
    • 2. Effects on innate immune system
    • 2.1. SCFA
    • 2.2. MAMPs and PAMPs
    • etc.

Minor points:

  • Fig.1 could include a drawing of the level of pathway step, eg. draw lumen if butyrate comes from there and draw mucosal outside when barrier function is increased, and epithelial or supepithelial layer for other steps
  • line 117; testis is mentioned here - are they also linked to neurological diseases, or BBB only? delete "commonly" here ("often" is sufficient)
  • Fig2: size needs to be increased
  • some typos throughout MS; paragraph starting line 229 is in Italic
  • line 427: delete "certainly" - we can't be sure yet
  • Abbreviations list needs to be re-formated

Author Response

I am not quite sure whether this article topic fits into the scope of "Vaccines" Journal
Suggest including a table summarizing the microbial molecules, their effects on immune system and related pathways, and the respective referencesResponse: We do not believe that the addition of such a table would help in the understanding of the content of this review since summarizing the interconnected pathways would not make for a clear and concise table.

Divide the paper clearly into "innate" and "adaptive" aspects, and put microbial components as sub-heading; e.g.
1. Intro
2. Effects on innate immune system
2.1. SCFA
2.2. MAMPs and PAMPs
etc.

The numbers were added to the beginning of each section of the text.

Minor points:

Fig.1 could include a drawing of the level of pathway step, eg. draw lumen if butyrate comes from there and draw mucosal outside when barrier function is increased, and epithelial or supepithelial layer for other steps Response: adding these to the figure would not add any more information to the figure and could even make it harder to understand. Therefore the figure was kept as is.
line 117; testis is mentioned here - are they also linked to neurological diseases, or BBB only? delete "commonly" here ("often" is sufficient) Response: “commonly” was removed and “linked to” was changed to “found in patients suffering from”. Because defaults in the BTB are found in some patients suffering from neurological diseases.
Fig2: size needs to be increased Response: Figure size was increased
some typos throughout MS;

paragraph starting line 229 is in Italic Response: I could not find the mentioned italic. It was potentially fixed by the editor.
line 427: delete "certainly" - we can't be sure yet Response:“certainly” was removed
Abbreviations list needs to be re-formated Response: I noticed that Vaccines does not have an abbreviations section therefore I removed it.

Reviewer 2 Report

The review Immune system modulations by products of the gut
 microbiota is a very good overview of the single active ingredients. The main bacteria that produce these metabolites have been named. The interactions gets much clearer.

I just have a few comments about the conntent:

 I am missing the link between nutrition - bacteria abundance and amount and diversity of metaboites. Please mention in some sentences.

Also the epigenetic impact of bacteria metabolites is missing. The interaction of how the immune system is regulated by the metabolites gets not clear in all cases. Only histone acetylation in section how butyrate regulates treg expression. And the impact of the expression levels due Tryptophan. Please add more mechanistic regulations to the other sections.

Paragraph 230 to 241 should not be italic.

Author Response

Comments and Suggestions for Authors
The review Immune system modulations by products of the gut
 microbiota is a very good overview of the single active ingredients. The main bacteria that produce these metabolites have been named. The interactions gets much clearer.

I just have a few comments about the conntent:

 I am missing the link between nutrition - bacteria abundance and amount and diversity of metaboites. Please mention in some sentences. Response: Mention of the effect of nutrition on the microbiota was added to lines 63-65

Also the epigenetic impact of bacteria metabolites is missing. The interaction of how the immune system is regulated by the metabolites gets not clear in all cases. Only histone acetylation in section how butyrate regulates treg expression. And the impact of the expression levels due Tryptophan. Please add more mechanistic regulations to the other sections. Response: We do not believe detailed information on epigenetic modifications are in the scope of this review.

Paragraph 230 to 241 should not be italic. Response: I could not find the mentioned italic. It was potentially fixed by the editor.

Reviewer 3 Report

The authors review recent literature on the interactions between gut microbiota and the immune system. Specifically, the authors describe the effect of various microbial metabolites on the activation of the immune system. The review delves into precise molecular mechanisms involved in the activation or inhibition of immune responses. The major activation pathways are discussed. The paper is well-written and well-structured. This paper will encourage further investigation of the dysbiosis and help developing new therapies.

In my opinion, this review needs a part describing different families of receptors pattern-recognition receptors. Also, I would include a short, broad review of TLR and NLRs.

Author Response

Comments and Suggestions for Authors
The authors review recent literature on the interactions between gut microbiota and the immune system. Specifically, the authors describe the effect of various microbial metabolites on the activation of the immune system. The review delves into precise molecular mechanisms involved in the activation or inhibition of immune responses. The major activation pathways are discussed. The paper is well-written and well-structured. This paper will encourage further investigation of the dysbiosis and help developing new therapies.

In my opinion, this review needs a part describing different families of receptors pattern-recognition receptors. Also, I would include a short, broad review of TLR and NLRs. Response: We added a small description of PRRs from line 76 to 86. The specific effects of each relevant PRRs are mentioned in the appropriate sections of the review.